# A New Gal in Town: A Systematic Review of the Role of Galanin and Its Receptors in Experimental Pain

**DOI:** 10.3390/cells11050839

**Published:** 2022-03-01

**Authors:** Diana Fonseca-Rodrigues, Armando Almeida, Filipa Pinto-Ribeiro

**Affiliations:** 1Life and Health Sciences Research Institute (ICVS), Campus of Gualtar, School of Medicine, University of Minho, 4710-057 Braga, Portugal; id10192@alunos.uminho.pt (D.F.-R.); aalmeida@med.uminho.pt (A.A.); 2ICVS/3B’s—PT Government Associate Laboratory, 4805-017 Guimarães, Portugal

**Keywords:** galanin, galanin receptors, pain, nociception

## Abstract

Galanin is a neuropeptide expressed in a small percentage of sensory neurons of the dorsal root ganglia and the superficial lamina of the dorsal horn of the spinal cord. In this work, we systematically reviewed the literature regarding the role of galanin and its receptors in nociception at the spinal and supraspinal levels, as well as in chronic pain conditions. The literature search was performed in PubMed, Web of Science, Scopus, ScienceDirect, OVID, TRIP, and EMBASE using “Galanin” AND “pain” as keywords. Of the 1379 papers that were retrieved in the initial search, we included a total of 141 papers in this review. Using the ARRIVE guidelines, we verified that 89.1% of the works were of good or moderate quality. Galanin shows a differential role in pain, depending on the pain state, site of action, and concentration. Under normal settings, galanin can modulate nociceptive processing through both a pro- and anti-nociceptive action, in a dose-dependent manner. This peptide also plays a key role in chronic pain conditions and its antinociceptive action at both a spinal and supraspinal level is enhanced, reducing animals’ hypersensitivity to both mechanical and thermal stimulation. Our results highlight galanin and its receptors as potential therapeutic targets in pain conditions.

## 1. Introduction

Chronic pain results from the abnormal function of the nervous system, in which pain persists beyond healing time (more than 3–6 months) [1], and affects approximately 20% of the European population [2,3]. This abnormal neuronal activity includes the sensitisation of the peripheral and the central nervous systems [4], leading to a heightened perception of pain [5]. However, the molecular mechanisms underlying the maintenance and development of chronic pain are still not fully clarified.

Galanin is a neuroendocrine 29-aminoacid neuropeptide (30 in humans), which was initially discovered by Tatemoto and colleagues [6]. Since then, galanin has been shown to play a key role in several physiological processes such as cognition [7], feeding [8], and nociception [9]. 

Under normal circumstances, this peptide occurs mostly in the dorsal root ganglia (DRG) [10] and the superficial layers of the spinal dorsal horn [11]. Importantly, it is present in a small population of primary sensory neurons [12,13] that give rise to small diameter fibres shown to co-express the neuropeptides calcitonin gene-related peptide (CGRP) and substance P (SP) [10,14,15]. The role of galanin in nociceptive modulation has been intensively investigated using different behavioural and electrophysiological techniques. Under normal conditions, galanin is thought to play a minor role in nociception, however it increases after injury, particularly in the DRG and spinal cord in which it plays a mostly antinociceptive role [16]. In this work, we systematically reviewed the literature regarding the role of galanin and its receptors in nociception at the spinal and supraspinal levels, as well as in the development and maintenance of chronic pain. 

## 2. Materials and Methods

### 2.1. Study Design 

This systemic review was conducted according to the Preferred Reporting Items for Systematic Reviews and Meta-Analyses (PRISMA) guidelines [17]. This review’s protocol was not registered prior to its submission.

### 2.2. Search Strategies

The literature search was conducted using the following electronic databases: PubMed, Web of Science, Scopus, ScienceDirect, OVID, TRIP and EMBASE. For each database, a combination of search criteria was used: “Galanin” AND “pain” from inception up to 26 July 2021. 

### 2.3. Eligibility Criteria 

Original articles that evaluated the role of galanin and its receptors in pain were included and divided into two different categories: (i) the role of galanin in nociceptive processing and (iii) changes in the expression of galanin and its receptors in different pain conditions and their role in pain development and maintenance. There was no restriction on the type of pain included in animal models. Publications were excluded when one of the following conditions was fulfilled: (i) non-original publications (reviews and book chapters); (ii) grey literature (conference abstracts, notes, letters to the editor); (iii) studies performed in human subjects; (iv) manuscripts written in languages other than English; and (v) manuscripts dealing with subjects other than pain and galanin (out of the scope).

### 2.4. Study Selection

After removing duplicates, two researchers (DFR and FPR) independently screened the title and abstract of every citation found in the literature search. In a second stage, full-text papers were screened against the inclusion/exclusion criteria. To qualify for inclusion, both investigators had to reach an agreement. A third investigator was involved in the case of unsolved disagreement (AA).

### 2.5. Data Extraction, Management, and Synthesis

Two researchers conducted the data extraction individually (DFR and FPR), which was posteriorly merged, and the discrepancies in the data extraction were all resolved by consensus. The following data, when available, were extracted from the included studies and compiled in a Microsoft Excel spreadsheet by all authors, including the following information: author, year of publication, species/strain of animals, type of pain studied, animal model, and the results regarding galanin and its receptors. Data from the included studies was synthesised in two tables and a narrative summary of the data is presented in the Results section. 

### 2.6. Quality Assessment

Two of the review authors independently assessed the quality of the studies using the ARRIVE guidelines [18], which comprises 21 criteria including study design, sample size, inclusion and exclusion criteria, randomisation in the allocation to experimental units, blinding during the study, outcome measures, statistical methods, description of species and the developmental stage of the animals, reporting of results, scientific background, study objectives, housing conditions, animal care and monitoring, scientific implications of the study, generalisability to other species, protocol registration, ethical approval statement and declaration of interest.

## 3. Results

### 3.1. Search Results

A total of 1379 publication references were initially retrieved using the above-mentioned search strategy (Figure 1) and one additional study was included from the reference lists of included studies using the snowball technique. After the removal of 887 duplicates, 492 publications were screened based on titles and abstracts and 460 publications were identified as potentially eligible. After screening the full text, 141 publications were included herein.

### 3.2. Characteristics of the Included Studies

A total of 36 articles were included that assessed the role of galanin and its receptors in nociceptive processing in different locations—the spinal cord and peripheral nerves, arcuate nucleus of the hypothalamus (ARC), the central nucleus of the amygdala (CeA), lateral habenula (LHb), nucleus accumbens (NAc) and periaqueductal grey (PAG)—and procedures such as intracerebroventricular administration and galanin overexpression/knockout. 

Regarding the role and changes in galanin and its receptors in different animal models of pain, a total of 105 articles were evaluated. While most studies were performed in rodents (97%), mostly rats (70%), three were performed in monkeys (Macaca mulatta). Different pain models were employed, mostly animal models of inflammatory (24%) and neuropathic pain (77%), such as carrageenan-induced inflammation and peripheral nerve injury, respectively. 

### 3.3. Assessment of Quality

Two of the review authors (MJP and FPR) independently assessed the quality of the reporting of the included studies, using the ARRIVE guidelines, as mentioned above. The potential range of the ARRIVE quality score was 0–20 and the overall mean score for the methodological quality of the studies included was 12.4 ± 2.2. A score between 0–1 was attributed to each criterion, and the mean score was calculated for each study. A global rating of strong was attributed to studies with a mean score higher than 15 (28.4%), moderate for a score between 10–15 (60.3%) and weak for those that scored under 10 (11.3%). Overall, the quality of the studies was good. The proportion of studies that met each criterion is summarised in the Appendix A.

## 4. The Role of Galanin in Pain Processing at the Spinal and Supraspinal Levels

The role of galanin in pain processing at the spinal levels has been the subject of many studies since its discovery, in which both endogenous and exogenous galanin are implicated (Table 1 and Appendix A).

Wiesenfeld-Hallin and colleagues first demonstrated that intrathecal administration of the putative galanin receptor antagonist M35 (galanin-(1–13)-bradykinin-(2–9)-amide) potentiated the facilitation of the flexor reflex [19]. When applied to peripheral nerves such as the saphenous and lumbar splanchnic nerves, galanin was shown to inhibit their response to noxious stimulation [20,21]. Likewise, galanin-over-expressing mice were shown to display reduced facilitation of the flexor reflex after C-fibre stimulation [22], as well as an increased thermal and mechanical nociceptive threshold [23]. Together, these results indicate that under normal conditions endogenous galanin plays a suppressing role in nociceptive processing in the spinal cord. However, the mechanisms associated with galanin’s inhibitory effect remain unclear, although much evidence points to a postsynaptic action of galanin in the dorsal horn [15], at least in part, mediated by protein kinase C (PKC) [24], and targeting primary afferent terminals to increase the release of substance P (SP) [25] and oxytocin [26]. 

The predominant effect of exogenous galanin on nociception is inhibitory, with several behaviour studies showing its intrathecal administration increases the response threshold to both mechanical, thermal, and inflammatory stimulation [25,27,28,29,30]. Importantly, exogenous galanin applied at the spinal level displays a biphasic effect upon nociception, potentially acting on primary afferents pre- or post-synaptically. The intrathecal administration of galanin at low doses [31] facilitates the spinal nociceptive flexor effect induced by C-fibre stimulation, and mechanically evoked thresholds of nociceptive afferents [32]. However, at higher doses, galanin produces a dose-dependent inhibition effect, and consequently antinociception, on these same afferents in both mice and rats [32,33,34,35]. Such a differential effect of galanin has been suggested to result from activation of different subtypes of receptors, which will be further discussed below. 

Galanin and its receptors are also important mediators of nociception and opiate-induced analgesia at the supraspinal level, as they are localised in important nociception-related structures such as the arcuate nucleus of the hypothalamus, the raphe nuclei, the striatum, the ventral hippocampus, and the locus coeruleus [13,36]. Accordingly, several electrophysiological and behavioural studies have been conducted to elucidate the role of galanin in pain modulation in the brain. In the studies included in this review, galanin showed an antinociceptive effect on the response to mechanical and/or thermal stimulation in healthy mice and rats when administered intracerebroventricularly (ICV) [24,37,38,39]; to the periaqueductal grey matter (PAG), a brain region involved in descending pain modulation [40,41,42]; to the arcuate nucleus of the hypothalamus (ARC), the major source of β-endorphin in the brain [43,44]; to the nucleus accumbens (NAc), a key structure in modulating rewards and pleasure processing [45]; to the lateral habenular nucleus (LHb), involved in pain-associated depression [46]; and to the central nucleus of the amygdala (CeA), which is primarily associated with emotional processing [47,48]. Additionally, when administered ICV, galanin was shown to display an antinociceptive effect on the trigemino-hypoglossal reflex [49,50]. Galanin’s antinociceptive effect was shown to be modulated by PKC, particularly in the CeA of rodents [51].

## 5. Galanin Expression and Modulation in Chronic Pain Models

Galanin plays an inhibitory role in spinal nociception and this role may be enhanced after peripheral nerve injury or inflammation (Table 2 and Appendix A, and Figure 2). Indeed, peripheral nerve transection has been shown to dramatically increase galanin expression in primary sensory neurons and their terminals in the spinal cord, which may act to reduce injury-induced hyperalgesia. Additionally, recent results further point to galanin as a key modulator of nociceptive processing at a supraspinal level, acting on several areas involved not only in the sensory-discriminative aspects but also on motivational-affective responses to pain.

### 5.1. Endogenous Galanin

Endogenous galanin plays a critical role in the development of hyperalgesia following peripheral injury, and as such is required for the development of peripheral and central sensitisation. This nociceptive role has been further confirmed by the development of knockout and transgenic animals. Galanin knockout animals are hyper-responsive to noxious stimulation after both carrageenan-induced inflammation [142] and peripheral nerve injury [75,107]. The opposite is observed in galanin-over-expressing mice, which display decreased hyperalgesia/allodynic responses and increased recovery after peripheral nerve injury [94,95]. As well, galanin administered systemically was able to reduce mechanical and thermal hyperalgesia (cold) in both inflammation and peripheral injury animal models [101]. 

### 5.2. Primary Sensory Neurons and Dorsal Root Ganglia (DRG)

In primary sensory neurons, galanin levels increase from non-detectable levels after chronic constriction injury of the sciatic nerve [61]. However, this protective mechanism does not appear to be based on changes in galanin expression in injured primary neurons [100]. A possibility is that nerve injury interrupts the anterograde transport of neuropeptides from the DRGs to the spinal cord, causing an accumulation of galanin.

The administration of galanin into the receptive fields of sensory fibres inhibited the response of dorsal horn neurons after spinal nerve ligation [113]. Additionally, after axotomy of the sciatic nerve, blocking galanin action by the administration of M-35 [99], a potent galanin antagonist, or antisense nucleotides [143] to the injured nerve potentiated the flexor reflex excitability [106] and increased autotomy behaviour in axotomised animals, suggesting a protective role of galanin after peripheral nerve injury. Similarly, the intra-articular administration of the antagonist M-35 doubled the responses to noxious stimuli in rats with kaolin/carrageenan-induced inflammation [144]. However, at low doses, the intra-plantar administration of galanin displayed a pronociceptive effect in capsaicin-evoked inflammatory pain, mediated by the activation of GalR2 receptors [102,104], and consequently, the PKC intracellular signalling pathway [103]. 

After peripheral axotomy, galanin is dramatically upregulated and expressed in small-sized DRG neurons, with a shift towards its expression also in medium/large-sized neurons [10,51,53,54,55,56,57,58,96,111,145], both of which are accompanied by a retrograde increase in galanin in the spinal cord. Similar results were obtained after partial sciatic nerve ligation [59,145,146], cisplatin-induced neuronopathy [64], spinal nerve ligation [72], constriction/photochemically-induced sciatic nerve injury [62,63,65,66,67,71,76], in inferior alveolar neuromas [68,69], tibial nerve injury [51,77,147], diabetes-induced neuropathy [76], medial nerve injury [78], trigeminal nerve injury [70], Freud’s adjuvant-induced inflammation [79,145], collagen-induced arthritis [80], bone cancer [73], post-herpetic neuralgia [74] and HIV-associated neuropathic pain [148]. 

This increase in galanin levels could be possibly due to an augmented galanin synthesis or decreased release, causing its accumulation in the somata of sensory neurons. Interestingly, a study by Ma and Bisby [56] showed this increase is significantly higher in constriction and partial nerve transection models, in comparison with total axotomy of the sciatic nerve. Conceivably, as during both chronic constriction and partial nerve transection, the surviving axons share an environment where adjacent axons are undergoing Wallerian degeneration, this setting might modulate galanin expression in both spared and axotomised DRG neurons. Indeed, different molecules such as leukaemia inhibitory factor (LIF) [55], acidic and basic fibroblast growth factor (aFGF, bFGF), nerve growth factor (NGF) [54], glial cell line-derived neurotrophic factor (GDNF) [72] and glial nuclear factor kappa B (NF-κB) [67] were shown to be key modulators of severe galanin upregulation observed after nerve injury. 

### 5.3. Spinal Dorsal Horn (SDH)

The results in the literature are contradictory in regard to the SDH. An increase of galanin levels in the spinal cord was detected in spinal nerve ligation and injury models [72,86], particularly in glial cells and their processes [83], as well as in small and medium-sized neurons of the dorsal horn with increased branching after axotomy [51,60,149]. Similarly, after noxious colorectal distension (CRD), the spinal levels of galanin increased gradually and peaked after 24 h [14]. After chronic constriction injury of the sciatic nerve, stimulation of the nerve was shown to increase galanin immunoreactivity in the spinal cord, which originated in primary afferent neurons [84]. However, this effect was not persistent and a decrease in galanin was observed in the weeks following injury [87]. Interestingly, increased levels of galanin were observed in the gracile nuclei after sciatic nerve injury [53,56,82], suggesting that some of the inhibitory effects of galanin may be mediated through the gracile nucleus pathway. The opposite was observed immediately after Freud’s adjuvant-induced inflammation, as galanin levels initially decreased [81,88], but as inflammation progressed, its content gradually increased, and galanin levels normalised [88]. A possible explanation for the initial decrease could be that noxious stimulation depletes the cellular stores of releasable galanin, and consequently, basal levels decrease below those present before any stimulation. 

Galanin administered at the spinal level inhibited the activity of wide-dynamic-range neurons in a dose-dependent manner [110] and the response of SDH neurons to mechanical, thermal and electrical stimulation after spinal nerve ligation [112]. Similarly, in chronic pain animal models, intrathecal galanin decreased mechanical and thermal hyperalgesia after carrageenan-induced inflammation [105], kaolin/carrageenan-induced arthritis [144], chronic constriction injury of the sciatic nerve [109,150], photochemically-induced nerve injury [108] and sciatic nerve-pinch injury [16,76] and diabetes-induced neuropathy [76]. Additionally, spinal administration of galanin reduced the endogenous levels of galanin in the DRGs and the dorsal horn after sciatic nerve pinch and diabetes-induced neuropathy [76]. These data suggest an antinociceptive role of galanin, which together with the fact that galantide administration increased the activity of wide-dynamic-range neurons [110], explains the increased autotomy observed after administration of galanin [99,106,143].

### 5.4. Supraspinal Galanin

An antinociceptive role of exogenous galanin has been recently demonstrated at the supraspinal level, particularly in the ARC and PAG. An increase in galanin expression in the ARC after nerve injury has been demonstrated in spared nerve injury [89] and cyclophosphamide (CP)-induced cystitis [90] animal models. Additionally, the administration of galanin to this area reduced mechanical and thermal hyperalgesia after carrageenan-induced inflammation [135] and chronic constriction injury of the sciatic nerve [136], with an increase in the number of galaninergic neurons. Galanin administration to the PAG also had an analgesic effect after chronic constriction injury of the sciatic nerve [134]. Indeed, galanin was shown to activate the beta-endorphinergic pathway from the ARC to the PAG [50], resulting in increased levels of serotonin [151] and galanin [90] in the medulla. Accordingly, it has been proposed that galanin regulates the excitability of the β-endorphinergic neurons in the ARC, and thereby the release of β-endorphin in the PAG [50]. Moreover, as most of the ARC neurons are projecting neurons, it is also possible that exogenous galanin acts through galaninergic projecting neurons. 

Galanin was also shown to be involved in pain modulation in several other brain areas, as the administration of galanin reduced mechanical and thermal hyperalgesia when administered to (i) the medulla after spared nerve injury [115]; (ii) the NAc, after carrageenan-induced inflammation [137] and chronic constriction injury of the sciatic nerve [92] in a dose-dependent manner, an effect blocked by galantide [92]; (iii) the CeA, after chronic constriction injury of the sciatic nerve [131]; (iv) the ACC after carrageenan-induced inflammation, in which an increased expression of galanin was observed [138]; and (v) the tuberomammillary nucleus of the hypothalamus after inflammation and chronic constriction injury of the sciatic nerve [132], an effect also blocked by galantide. Additionally, the subarachnoid transplantation of immortalised galanin-over-expressing astrocytes was shown to reduce mechanical and thermal hyperalgesia after spared nerve injury [139]. 

## 6. Receptor Mechanisms Underlying the Varying Roles of Galanin

Galanin acts through three main receptors: galanin receptor 1 (GalR1), galanin receptor 2 (GalR2) and galanin receptor 3 (GalR3), all belonging to the family of G-protein coupled receptors (GPCR). These galanin receptors are differentially distributed, and all three are present in DRGs and the spinal cord [152]. 

The first known galanin receptor, GalR1, is the most abundant and widespread in the CNS of adult rodents [153], and GalR1 mRNA was found in many brain areas such as the hippocampus, amygdala, ventral tegmental area (VTA) and the NAc [154]. GalR1 receptors are located predominantly post-synaptically [21], and the presence of GalR1 in DRG, the SDH and primary afferents suggest an antinociceptive effect upon primary afferent terminal excitability [27,29]. Indeed, the inactivation [24], reduction in GalR1 levels [41], or GalR1-knockout [121,122] causes a partial blockade of the inhibitory effect of galanin, increasing the mechanical and thermal hypersensitivity. The selective destruction of GalR1-expressing neurons in the SDH also reduced thermal sensitivity to heat (46). GalR1 is also an important player in galanin-induced antinociceptive effects in the brain. Accordingly, the administration of M617, a selective agonist to GalR1 [155], was shown to cause a decrease in the sensitivity to both thermal and mechanical stimulation when administrated intracerebroventricularly [45], to the CeA [25,128,131] and the PAG of rats [26]. The activation of GalR1 was further shown to inhibit protein kinase A (PKA), particularly in the CeA of rodents, with an antinociceptive role after nerve injury [123]. These results further support the notion that galanin induces antinociception in rodents through activation of GalR1 receptors at both the spinal and supraspinal level. 

GalR2 is mostly expressed in DRGs and brain areas such as the hippocampus, cerebellar cortex, hypothalamus, and amygdala [156]. GalR2 receptors are localised both pre- and post-synaptically [21]. The activation of GalR2 suppresses Ca^2+^ channel currents [21], and consequently increases the content of Ca^2+^ [153], which could further activate CAMKII [130] and MAPK [157], thus modulating nociception and neuroplasticity. Additionally, GalR2 activation was shown to enhance NPYY1R-mediated signalling [158], which leads to increased anxiety-like behaviours, and could possibly be altered in pain modulatory areas. These receptors are associated with a more pronociceptive action of galanin, as the selective destruction of GalR2-expressing neurons causes the loss of a subset of sensory neurons in the DRG (likely nociceptors) and reduces neuropathic and inflammatory pain responses [27]. This phenomenon was also observed after axotomy, although with no apparent impact on the mechanical and thermal nociceptive responses [125]. Further evidence of a pronociceptive role of GalR2 was demonstrated when intraplantar administration of GalR2 agonist M1896 increased mechanical and thermal nociceptive responses after chronic constriction injury of the median nerve, with GalR2 antagonist M871 having an opposite effect [78]. However, different results were obtained after administration of a GalR2-preferring galanin analogue, which displayed an analgesic effect after carrageenan-induced induction and partial sciatic nerve ligation [127]. There are few reports regarding the role of GalR2 in pain modulation in the brain. Nonetheless, GalR2 was shown to be involved in galanin-induced antinociception at the supraspinal level, as the administration of M871 to the PAG [48], the NAc [129,130] and the ACC [138] attenuated the antinociceptive effects of galanin. 

The distribution of GalR3 in the central nervous system is rather restricted when compared to GalR1 and GalR2, being present primarily in the preoptic/hypothalamic area [159]. Although the mechanisms of GalR3 are still largely unknown, this receptor is expressed on murine neutrophils and has been shown to influence the vascular components of inflammatory processes. Indeed, recent studies showed that GalR3 knockout animals display increased disease severity and oedema after autoimmune arthritis [119], suggesting a mainly anti-inflammatory role for this receptor. 

After inflammation or nerve injury, galanin receptors GalR1 and GalR2 show great expression plasticity at the spinal, the DRG, and the supraspinal levels. After peripheral nerve injury and inflammation, GalR1 was downregulated [16,76,120] while GalR2 was upregulated [16,76,78,118,124,126] in both the DRG and SDH. These results are consistent with a more pronociceptive action of GalR2 in the spinal cord, causing hypersensitivity to noxious stimulation and the consequent mechanical and thermal hyperalgesia observed in these animals. Yet, in spinal cord injury [86], trigeminal nerve injury [70] and streptozotocin-induced diabetes [116], GalR2 expression in DRG decreased as well as both receptors expression in the dorsal horn, suggesting a different mechanistic action of galanin and its receptors in these pathologies. At the supraspinal level, an increased expression of both GalR1 and GalR2 receptors was observed in the tuberomammillary nucleus (TM) [132], the NAc [128,137] and the ACC [138] in inflammatory and nerve injury animal models. 

The differences between receptor subtypes contribute to the diversity of the possible physiological effects and the pharmacological relevance of galanin in nociception and pain. It was proposed that galanin displays a biphasic and dose-dependent effect on nociception, through the action of inhibitory (antinociceptive) GalR1 receptors or excitatory (pronociceptive) GalR2 receptors [82,160]. Indeed, a low dose of galanin has a pronociceptive role at the spinal level, which was shown to be mediated by GalR2 receptors, whereas at higher doses, the antinociceptive role of galanin is mediated by GalR1 [82]. Different mechanisms have been suggested for this dual role of galanin receptors in nociception. In the substantia gelatinosa (SG), galanin at lower concentrations was shown to enhance the release of L-glutamate from nerve terminals onto SG neurons by activating GalR2/R3, whereas galanin at higher concentrations produced membrane hyperpolarisation by activating GalR1 [22].

## 7. Other Mechanisms

Considering the direct antinociceptive effect of opioid analgesics, the clinical application of epidural or intrathecal morphine for pain relief is a common medical procedure. However, morphine is a potent inducer of tolerance and dependence. To reduce the effective dose of opioids and their consequent adverse effects, different attempts have been made to combine morphine with other receptor agonists such as galanin. 

Galanin can act synergistically with opiates to suppress spinal hyperexcitability, being a potential target for the management of patients suffering from chronic pain, particularly when combined with morphine. Intrathecal administration of galanin reduced the morphine dose required for the suppression of the flexor reflex [97], and the further administration of the galanin antagonists’ galantide and M-35 almost completely abolished the antinociceptive effect of morphine [31]. This interaction is mediated mostly by mu-opioid receptors [141], which are significantly increased after nerve injury [161]. Accordingly, after binding to mu-opioid receptors, galanin may facilitate the inhibitory effects of opioid peptides, and/or enhance their affinity with their receptors. The administration of galanin receptor antagonists prevents galanin from binding to its receptors and indirectly attenuates the analgesic effect of opioids. 

Galanin was also shown to interact with opioid actions in the brain, particularly in the CeA, ARC and the PAG. The injection of galanin ICV [42] or directly into these areas [34,134] potentiated the action of morphine, and its antinociceptive effect was reversed by opioid receptor antagonists, namely, non-selective naloxone and µ-selective β-FNA [33,47]. This interaction between galanin and opioid receptors occurs via the neural pathway from the ARC to the PAG [33] and is mediated by the excitatory GalR2 receptors [47].

## 8. Conclusions

This review not only indicates that galanin has a mostly antinociceptive role at both the spinal and supraspinal levels, but also suggests a possible interaction between galanin and the endogenous opioid system. Since chronic pain could develop from insufficient galaninergic control of nociception, targeting galanin receptors (particularly GalR1) could therefore be a potential therapeutic strategy, especially when paired with opioids.

## Figures and Tables

**Figure 1 cells-11-00839-f001:**
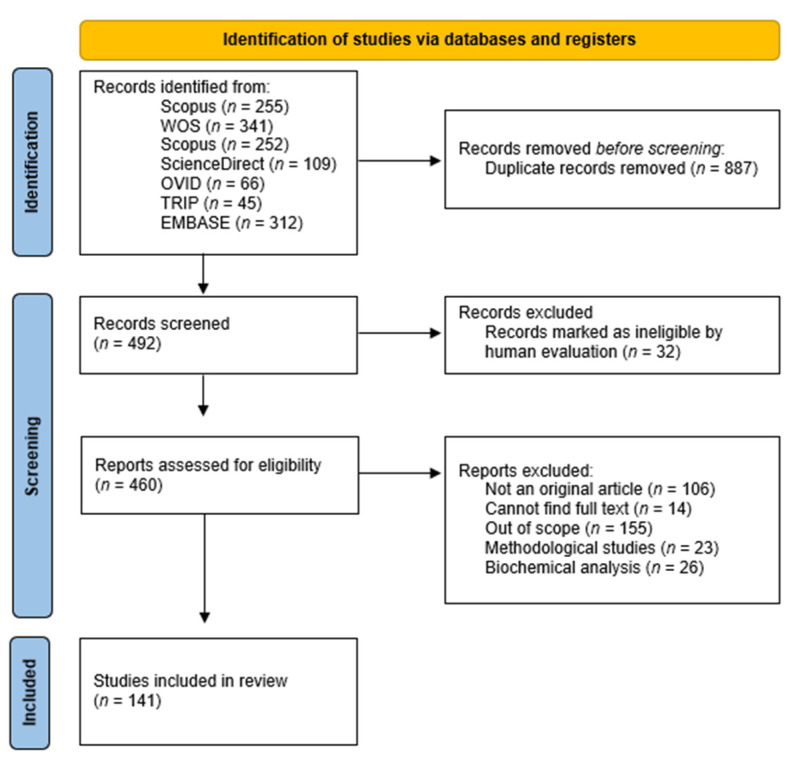
Flowchart representation of the different stages of the selection process.

**Figure 2 cells-11-00839-f002:**
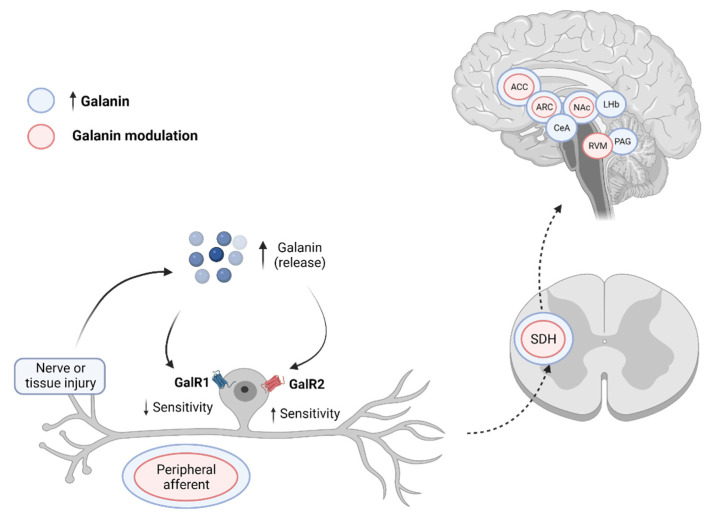
Mechanism of action of galanin in chronic pain conditions. Under normal situations, peripheral nerve injury or inflammation induces an increase in galanin levels that act upon GalR1 receptors, inhibiting nociceptive transmission. In chronic pain conditions, there is an upregulation of GalR2 receptors in peripheral nerve afferents and their correspondent dorsal root ganglia (DRG), which causes hypersensitivity to noxious stimulation and consequent mechanical and thermal hyperalgesia. Peripheral injury was also shown to increase galanin levels at both a spinal and supraspinal level, and its administration produces an antinociceptive effect upon nociceptive processing. (ACC—anterior cingulate cortex; ARC—arcuate nucleus of the hypothalamus; CeA—central nucleus of the amygdala; LHb—lateral habenula; NAc—nucleus accumbens; PAG—periaqueductal gray; RVM—rostral ventromedial medulla; SDH—spinal dorsal horn). Created in BioRender.com.

**Table 1 cells-11-00839-t001:** Summary of the literature search on the role of galanin and its receptors in the processing of nociceptive information.

Mechanism	Effects
Endogenous Galanin	Increase in sensory neurons after administration of resiniferatoxin, an ultrapotent capsaicin analog [18]
Galanin Overexpression	Antinociceptive effect on thermal and mechanical sensitivity [19]
	Reduced facilitation of the nociceptive flexor reflex [20]
Galanin Receptors	GalR1 receptors are located predominantly post-synaptically whereas GalR2 receptors may be localised both pre- and post-synaptically in the spinal cord [21]
	Galanin at lower concentrations activates GalR2/R3, whereas galanin at higher concentrations also activates GalR1 [22]
	GalR1 activation, but not GalR2/3 activation, suppresses mechanical sensitivity [23]
	Inactivation of GalR1 attenuates the antinociceptive effect of galanin [24]
	GalR1 is an antinociceptive target in the central nucleus of the amygdala [25,26]
	Selective destruction of GalR1-expressing superficial dorsal horn neurons produces heat hypoalgesia [27]
	The absence of GalR2 induces the loss of a subset of sensory neurons (likely nociceptors) [28]
Interaction with Opioids	Potentiates the analgesic effect of morphine [29,30,31]
	Interaction between galanin and opioids [32]
	Galanin exerts its antinociceptive effects through the µ-opioid receptor [33]
	Both µ- and δ-opioid receptors are involved in galanin-induced antinociception [34]
Local Administration to Peripheral Nerves	Administration of galanin to the saphenous nerve truck inhibits axonal excitability (antinociceptive effect) [35]
	Administration of galanin to the lumbar splanchnic nerve reduces mechanical sensitivity (antinociceptive effect) [23]
Intrathecal Galanin	Antinociceptive effect on thermal and mechanical sensitivity [31,36,37,38]
	Antinociceptive effect on formalin-induced nociception [29]
	No effect on flexor reflex [30]
	Reduced facilitation of the nociceptive flexor reflex [24,39,40]
	Antinociceptive effect mediated by activation of spinal GalR1, but not GalR2 receptors [29]
	Antinociceptive effect mediated by activation of GalR2/3 receptors [41]
Supraspinal Galanin	Intracerebroventricular administration of galanin: - no effect on mechanical and thermal sensitivity after administration of N-terminal galanin fragment [42] - antinociceptive effect on thermal and mechanical sensitivity [43,44,45] - reduced facilitation of the nociceptive trigemino-hypoglossal reflex [33,46]
	Galanin administration to the periaqueductal grey (PAG) has an antinociceptive effect on thermal and mechanical sensitivity [32,47,48] Activation of GalR1 induces antinociception in rats with morphine tolerance [47] GalR2 antagonist administration (M871) attenuates the antinociceptive effects of galanin [48]
	Galanin administration to the arcuate nucleus of the hypothalamus (ARC)—decreases thermal and mechanical sensitivity [49,50]
	Galanin administration to the central nucleus of the amygdala (AMY)—decreases thermal and mechanical sensitivity [25,34]
	Galanin administration to the lateral habenula complex (LHb)—decreases thermal and mechanical sensitivity [26]

**Table 2 cells-11-00839-t002:** Summary of the literature search on the role of galanin and its receptors in different animal models of experimental pain. (ACC—anterior cingulate cortex; ARC—arcuate nucleus of the hypothalamus; CeA—central nucleus of the amygdala; DRG—dorsal root ganglia; NAc—nucleus accumbens, RVM—rostral ventromedial medulla, TM—tuberomammillary nucleus).

Mechanism	Effects
Galanin Levels—DRG	Increased in DRG neurons in animal models of neuropathic pain: sciatic nerve axotomy [10,15,51,52,53,54,55,56,57,58,59,60] chronic constriction injury of the sciatic nerve [56,61,62,63] cisplatin-induced neuropathy [64,65,66,67] alveolar nerve axotomy [68,69,70] photochemically induced sciatic nerve injury [71] spinal nerve ligation [72] sarcoma-induced cancer pain [73] varicella zoster virus-induced neuropathy [74] partial saphenous nerve ligation injury [75] sciatic nerve pinch [16,76] tibial nerve injury [77] median nerve chronic constriction injury [78]
	Increased in DRG neurons in animal models of inflammatory pain: CFA induced arthritis, especially at time-course points with high inflammation and severe joint destruction [79] collagen antibody-induced arthritis [80]
Galanin Levels—Spinal	Increased in the spinal cord in animal models of neuropathic pain: flexion of inflamed ankles [81] sciatic nerve axotomy [53,56,82] spinal nerve ligation [72,83] chronic constriction injury of the sciatic nerve [56,67,84,85] noxious colorectal distension [14] sciatic nerve pinch [16] streptozotocin-induced diabetes [76] spinal cord injury [86]
	Decreased in the spinal cord in animal models of neuropathic pain: cisplatin-induced neuropathy [64] chronic constriction injury of the sciatic nerve [87]
	Decreased in the spinal cord at the onset of Freud’s adjuvant-induced inflammation, which gradually increases [88]
Galanin Levels—Supraspinal	Increased in the ARC after: spared nerve injury [89] visceral pain induced by cyclophosphamide (CP) [90]
	Increase in the RVM and the dorsal raphe nucleus in monoarthritis [91]
	Increased in the NAc after chronic constriction injury of the sciatic nerve [92]
Effect of Galanin Knockout or Overexpression	Decreased in the spinal cord at the onset of Freud’s adjuvant-induced inflammation, which gradually increases [88]
	No change in spinal galanin levels after collagen antibody-induced arthritis [93]
	Galanin suppression increases allodynic responses after sciatic nerve axotomy [94]
	Galanin overexpression decreases thermal/mechanical hyperalgesia after sciatic nerve injury [94,95]
	Galanin over-expressing animals displayed increased levels of galanin in the DRG and their corresponding nerve terminals after sciatic nerve axotomy [96]
Role of Endogenous Galanin	Galanin had a biphasic effect on the flexor reflex in rats with intact nerves, including facilitation, followed by depression, in a dose-dependent manner [97]
	Intrathecal injections of antibodies against galanin inhibited carrageenan-induced hyperalgesia [98]
	M35 administration has a facilitatory effect on flexor reflex excitability, which was potentiated after nerve axotomy [99]
	M35 administration enhances autotomy behaviour after sciatic nerve axotomy [100]
	Intra-arterial infusion of galanin inhibits acetone and menthol responses in the naive rodent and following models of neuropathic (partial sciatic nerve injury) and inflammatory pain (carrageenan) [101]
	Intraplantar administration of galanin at low doses increases capsaicin-evoked nociceptive behaviours [102,103,104]
Intrathecal Administration of Galanin	Reduces carrageenan-induced inflammation and hyperalgesia [105]
	Reduced facilitation of the nociceptive flexor reflex after sciatic nerve axotomy [106]
	Low doses of galanin have a pronociceptive effect on mechanical and cold allodynia after chronic constriction injury of the sciatic nerve [107]
	Antinociceptive effect on mechanical/thermal hyperalgesia after: photochemically-induced sciatic nerve injury [108] chronic constriction injury of the sciatic nerve [109,110] kaolin/carrageenan-induced arthritis [111] spinal nerve ligation [112,113] carrageenan-induced inflammation [114] spared nerve injury [115] sciatic nerve-pinch injury [16,76] streptozotocin-induced diabetes [76,116]
	Altered the responses of mechano-nociceptive C-fibre afferents in a dose-dependent manner in both naive and nerve-injured animals, with low concentrations facilitating and high markedly inhibiting mechano-nociceptor activity [117]
Role of Galanin Receptors	Decreased expression of GalR1 after in DRG and spinal cord neurons: carrageenan-induced inflammation [118] sciatic nerve axotomy [118,119] streptozotocin-induced diabetes [76,116] spinal nerve ligation [120] sciatic nerve pinch injury [76]
	GalR1 knockout animals display increased mechanical and thermal hypersensitivity after sciatic nerve injury [121]
	GalR1 knockout mice have no differences concerning acute nociception but showed a modest tendency towards increased hyperalgesia after tissue injury and inflammation [122]
	Activation of GalR1 reduces CAP-induced inflammatory pain, while the opposite is observed after activation of GalR2 [104]
	The modulatory effects of galanin on cooling are independent of GalR2 and GalR3 activation but mediated by activation of GalR1 [101].
	Activation of GalR1, but not GalR2, attenuated diabetic neuropathic pain [116]
	GalR1 activation results in the inhibition of the PKA and induces antinociceptive effects after chronic constriction injury of the sciatic nerve [123]
	Increased expression of GalR2 in DRG and spinal cord neurons after: carrageenan-induced inflammation [118] sciatic nerve pinch injury [76] median nerve chronic constriction injury [78] spared nerve injury [124]
	Decreased expression of GalR2 in DRG and spinal cord neurons after: sciatic nerve axotomy [118,119] streptozotocin-induced diabetes [76,116] spinal cord injury [86] alveolar nerve axotomy [70]
	Lack of the GalR2 results in a considerable developmental loss of DRG neurons after spinal nerve injury [125] and sciatic nerve axotomy [126]
	Activation of GalR2 has an antinociceptive effect after nerve injury and inflammation [127]
	A low dose of galanin has a pronociceptive role at the spinal cord level, which is mediated by GalR2 receptors whereas the antiallodynic effect of high-dose galanin on neuropathic pain is mediated by the GalR1 receptors [107]
	Increased expression of GalR1 and GalR2 in the NAc after: carrageenan-induced inflammation [128] chronic constriction injury of the sciatic nerve [128,129]
	GalR2 activation in the NAc induces CAMKII and PKC after carrageenan-induced inflammation [130]
	Increased expression of GalR1 in the CeA after chronic constriction injury of the sciatic nerve [131]
	Increased expression of GalR1 in the TM after chronic constriction injury of the sciatic nerve [132]
	GalR2 is involved in the galanin-induced antinociception in the ACC [119,133]
	GalR3 does not mediate mechanical hyperalgesia in autoimmune arthritis [119]
Supraspinal Administration of Galanin	Galanin administration to the PAG decreases mechanical and thermal hyperalgesia after chronic constriction injury of the sciatic nerve [134]
	Galanin administration to the ARC decreases mechanical and thermal hyperalgesia after: carrageenan-induced inflammation [135] sciatic nerve ligation [136]
	Galanin administration to the TM decreases mechanical and thermal hyperalgesia after carrageenan-induced inflammation and chronic constriction injury of the sciatic nerve [132]
	Galanin administration of galanin to the dorsomedial hypothalamic nucleus is pronociceptive in awake healthy and kaolin/carrageenan-arthritic animals [91]
	Galanin administration to the NAc decreases mechanical and thermal hyperalgesia after: carrageenan-induced inflammation [137] chronic constriction injury of the sciatic nerve [92,128]
	Administration of M35 in the NAc attenuated the antinociceptive effects of galanin after chronic constriction injury of the sciatic nerve [123]
	Galanin administration to the ACC decreases mechanical/thermal hyperalgesia after: carrageenan-induced inflammation [138] chronic constriction injury of the sciatic nerve [133]
	Galanin administration to the CeA decreases mechanical/thermal hyperalgesia after chronic constriction injury of the sciatic nerve [131]
	Subarachnoid transplantation of immortalised galanin-over-expressing astrocytes has an antinociceptive effect after spared nerve injury [139]
Interaction with opioids	Galanin acts synergically with opioids to inhibit the nociceptive information transmission in animal models of chronic constriction injury of the sciatic nerve [134,140,141]

## Data Availability

The data underlying this article are available in the article and the online Appendix A.

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
