# Peer review of "A New Gal in Town: A Systematic Review of the Role of Galanin and Its Receptors in Experimental Pain"

_cells, 2022, doi:10.3390/cells11050839_

Round 1
Reviewer 1 Report
line 620 The IC50 for ASIC channel is wrong and the reference to it is not appropriate.
Ref 160 --> ~whereas IC50 for ibuprofen was 3.42 ± 0.50 mM (n= 9)
But, only the IC5 of this part is presented, neither is the IC50 for ASCI1a.
The IC50 values suggested by the authors can be found here in Table 1 of the journal.
Anne Baron, Eric Lingueglia. Neuropharmacology. 2015 ;94:19-35. Pharmacology of acid-sensing ion channels - Physiological and therapeutical perspectives
In order to correct such an error, a more detailed confirmation is required as a whole in addition to the pointed out part.
Author Response
I believe there has been an upload mistake, the reviewer’s comments are not related to our manuscript, as the authors were unable to find the suggested changes. If the reviewer is willing to resubmit the comments to our manuscript, we will do our best to accommodate the suggestions.
Reviewer 2 Report
|
Title: “A new Gal in town: a systematic review on the role of galanin and its receptors in experimental pain“ No: 1604398
A Reviewer comment: The manuscript describes the role of galanin and its receptors under pathological condition. The paper contains something new and potentially deserve for publication.
|
Author Response
The authors thank the reviewer for the comment.
Reviewer 3 Report
Overall, the review is well prepared and structured. The role of Galanin in pain is carefully detailed. However, to complete the description of the GALR2 actions is necessary to add some information. GALR2 could stimulate Ca2+ entry and CaMK, as pointed out, but also MAPK (Borroto-Escuela et al. J Cell Physiol. 2021 May;236(5):3565-3578. doi: 10.1002/jcp.30092) that could be relevant for plasticity actions on pain modulation. Moreover, GALR2 inhibits Gi/o and CREB response (Narváez et al., Brain Struct Funct. 2015 Jul;220(4):2289-301. doi: 10.1007/s00429-014-0788-7) that could be related to the analgesic effects in other cellular regions.
Author Response
We thank the reviewer for the suggestions and have completed the description of GalR2 actions accordingly. Therefore, Section 6 (Receptor mechanisms underlying the varying roles of galanin), now reads: “GalR2 is mostly expressed in DRGs and brain areas such as the hippocampus, cerebellar cortex, hypothalamus, and amygdala [156]. GalR2 receptors are localized both pre- and post-synaptically [21]. The activation of GalR2 suppresses Ca2+ channel currents [21] and consequently increasing in the content of Ca2+ [153], which could further activate CAMKII [130] and MAPK [157], thus modulating nociception and neuroplasticity. Additionally, GalR2 activation was shown to enhance NPYY1R-mediated signalling [158], which lead to increased anxiety-like behaviours, and could possibly be altered in pain modulatory areas.”